# Development and Physical Characterization of α-Glucan Nanoparticles

**DOI:** 10.3390/molecules25173807

**Published:** 2020-08-21

**Authors:** Kervin O. Evans, Christopher Skory, David L. Compton, Ryan Cormier, Gregory L. Côté, Sanghoon Kim, Michael Appell

**Affiliations:** 1Renewable Product Technology Research Unit, National Center for Agricultural Utilization Research, USDA, 1815 N, University Street, Peoria, IL 61604, USA; Chris.Skory@usda.gov (C.S.); David.Compton@usda.gov (D.L.C.); cormier.ryan@gmail.com (R.C.); cotegl@hotmail.com (G.L.C.); 2Plant Polymer Research Unit, National Center for Agricultural Utilization Research, USDA, 1815 N. University Street, Peoria, IL 61604, USA; sanghoon.kim@usda.gov; 3Mycotoxin Prevention and Applied Microbiology Research Unit, National Center for Agricultural Utilization Research, USDA, 1815 N. University Street, Peoria, IL 61604, USA; michael.appell@usda.gov

**Keywords:** sucrose polysaccharide, biodegradable, nanoparticles, zeta potential, pH effect

## Abstract

α-Glucans that were enzymatically synthesized from sucrose using glucansucrase cloned from *Leuconostoc mesenteroides* NRRL B-1118 were found to have a glass transition temperature of approximately 80 °C. Using high-pressure homogenization (~70 MPa), the α-glucans were converted into nanoparticles of ~120 nm in diameter with a surface potential of ~−3 mV. Fluorescence measurements using 1,6-diphenyl-1,3,5-hexatriene (DPH) indicate that the α-glucan nanoparticles have a hydrophobic core that remains intact from 10 to 85 °C. α-Glucan nanoparticles were found to be stable for over 220 days and able to form at three pH levels. Accelerated exposure measurements demonstrated that the α-glucan nanoparticles can endure exposure to elevated temperatures up to 60 °C for 6 h intervals.

## 1. Introduction

There is an ever-increasing drive to find biopolymers capable of forming nanoparticles. It is of special interest to find biopolymers that form nanoparticles capable of readily encapsulating hydrophobic bioactive agents because biopolymers tend to have better biocompatibility, better bioactivity, and/or lower toxicity than nanoparticles made from metals and/or synthetic polymers [1,2]. It is also important that the biopolymers suitable for making nanoparticles are biodegradable and come from a renewable source. Biodegradable biopolymers are likely to leave the least impact on the environment, and being from a renewable source means the ability to readily replenish the material.

α-Glucans are polysaccharides with a repeating d-glucose monomer that can be produced enzymatically using glucansucrase, which catalyzes the transfer of α-d-glucopyranosyl units from sucrose to acceptor molecules to form α-glucan chains. These enzymes are synthesized by numerous lactic acid bacteria and can form several different α-glucosidic linkages that affect the branching and water solubility of the polysaccharide. α-glucans, such as dextran, contain mostly α (1→6) linkages and are readily soluble in aqueous solvents. In contrast, α-glucans that have a significant portion of α (1→3) linkages often form extended ribbon-like helices, rendering the biopolymers insoluble [3,4,5]. Thus, the α-glucan polysaccharides studied in this work are natural, biodegradable, and highly renewable materials for making nanoparticles.

Water-insoluble glucans, which are produced by several species of *Streptococcus* and *Leuconostoc* bacteria [6,7,8,9,10,11,12], usually have a mixture of both linkage types, imparting an amphipathic characteristic to the polysaccharide that can be used to create novel nanoparticles. We previously showed that high-pressure homogenization (10–60 passes at 35 to 200 MPa) of water-insoluble glucans could be used to form nanoparticles with the potential for encapsulation or film formation. We showed that several different water-insoluble glucans could be used to form nanoparticles with varying size and polydispersity. Polysaccharide produced from a cloned glucansucrase from *Leuconostoc mesenteroides* NRRL B-1118 typically yielded the most reproducible nanoparticles with a diameter less than 200 nm. Physical characterization of these nanoparticles, such as surface charge or stability, has been limited. This work explores the thermal properties of α-glucans that are not nanoparticles, the stability of the α-glucan nanoparticles over a temperature range of 10 °C to 86 °C using fluorescence anisotropy, and the nanoparticles’ size and zeta potential over a 7-month period and under accelerated exposure at 23 °C, 37 °C, and 60 °C.

## 2. Results and Discussion

### 2.1. DSC Analysis

Thermal properties of the α-glucan polysaccharides were investigated using DSC measurements because heat gain and loss may be accompanied by structural changes within the polysaccharide.

Figure 1 shows the result of the thermal analysis of pure α-glucan polysaccharides prior to their formation into nanoparticles. Indications are that the native α-glucan polysaccharides exhibited a small endothermic peak near 70 °C. This is interpreted as the transition temperature for the start of structural changes in the α-glucans. The data suggest that structural unwinding is completed at approximately 80 °C, which may be the degradation temperature where nanoparticles made from these glucans completely unravel. It should be noted that a similar temperature range was found for exopolysaccharides produced from *Lactobacillus plantarum* KF5 isolated from Tibet kefir grains [13] but is well below the melting temperature characteristic of levan synthesized by levansucrase.

Further thermal analysis was conducted on α-glucan polysaccharides, prior to their being made into nanoparticles, by exploring TGA measurements; Figure 2 shows the results. Degradation for the α-glucan polysaccharides occurred in two stages. The first stage of weight loss, which occurred from 25 °C to about 100 °C, was similar to that found for chitosan [14] and accounted for approximately 9.2% of mass contents. This was likely due to the loss of weight from adsorbed water being evaporated. The second stage did not start until the temperature was ~160 °C and continued to ~340 °C, where 18–19% mass remained, which was likely due to the thermal breakdown of the α-glucan polysaccharides. Differential TGA shows that 50% of the α-glucan polysaccharides’ mass degraded at around 320 °C. The TGA data confirm that the lower transition temperature presented by DSC measurements is associated with structural changes in the glucan since the mass loss associated with polysaccharide degradation did not occur prior to about 160 °C.

Figure 3a shows the water-insoluble glucan dispersed in water before and after homogenization. The mixture was cloudy prior to homogenization and readily reached clarity after homogenization. It was determined that approximately 70 MPa and 60 passes were the conditions necessary to optimize nanoparticle size (Figure 3b). Optimized conditions resulted in nanoparticles in the size range of 108 to 128 nm in diameter.

### 2.2. Morphology Analysis of α-Glucans and Nanoparticles

Figure 4A shows the results of SEM images of freshly prepared α-glucan that had been extensively rinsed with water and air-dried. Air-drying resulted in a non-uniform distribution of the polysaccharide and the appearance of apparently agglomerated spheres. Figure 4B shows the results of the air-dried samples after freeze-drying at −20 °C. It is shown that the agglomerated spheres disappeared and were replaced by a fibrous network of the polysaccharide. Figure 4C shows that the freeze-dried polysaccharides assembled into spherical nanospheres after homogenization and were similar in size to that determined by DLS measurements.

It is believed that nanoparticle formation can be partially attributed to the nature of the α-(1→3)-linked and α-(1→6)-linked d-glucose units in the polysaccharide. Previous work done on a similar glucan produced by *Streptococcal mutans* described the branched α-(1→6)-linked portion of the biopolymer as being responsible for its adherence to the hydrophilic hydroxylapatite in teeth [4,5]. High-pressure homogenization of the α-glucan could force the rigid α-(1→3)-linked regions responsible for its insolubility into the center of a forming sphere while forcing the α-(1→6)-linked regions to the surface to maximize the hydrophilic interaction with the aqueous media.

### 2.3. Fluorescence Anisotropy Analysis 

The hydrophobicity of the α-glucans makes them insoluble in water. Detecting hydrophobicity in glucans after they are converted to nanoparticles is possible using the fluorescent probe DPH because DPH preferentially intercalates into hydrophobic regions in the aqueous environments where it is quenched [15]. The addition of DPH to formed nanoparticles resulted in the typical emission spectrum of DPH inside a hydrophobic environment [15,16], indicating that polarization and subsequently anisotropy were measurable. Anisotropy measurements are based on the ability of a fluorescent probe molecule to rotate about its environment and on detecting the difference between the horizontal and vertical components of the probe’s fluorescence emission signal during this rotation [17]. The more the probe molecule rotates, the smaller the difference becomes between the vertical and horizontal components of the emitted light. Figure 5 shows the anisotropy of DPH, from 10–86 °C, when DPH is intercalated into α-glucan nanoparticles. DPH anisotropy with the nanoparticles exhibited its highest value (~0.625) from 10 to 18 °C, indicating the least rotational freedom of DPH. This suggests that molecular packing is tightest within the nanoparticles at 10–18 °C. Molecular packing within the nanoparticles decreased over the temperature range of 18 to 54 °C. Unexpectedly, the anisotropy started to increase above 55 °C and returned to near the previous level seen at 10–18 °C. This indicates that molecular packing increased and subsequently returned to previous levels, suggesting that the endothermic peak seen in Figure 1 is a transition temperature for the α-glucans, where the polysaccharide possibly unfolded. The fact that the anisotropy at 86 °C was about the same value as it was at 10 °C indicates that a significant population of nanoparticles did not fully unravel at high temperatures, suggesting that the nanoparticles are highly stable and resistant to temperature degradation up to nearly 90 °C. Additionally, the fact that DPH, which has a higher affinity for hydrophobic regions, was measurable fluorescently within these nanoparticles during the entire temperature range from 10 °C to 86 °C suggests that other hydrophobic bioactives will associate within the nanoparticles over the same temperature range (thus, binding kinetics and controlled-release studies will be studied over this temperature range in later work).

### 2.4. DLS and ζ-Potential Analysis 

Considering that the α-glucans homogenized into nanoparticles (Figure 3), it is important to characterize the stability of the nanoparticles. This was done by monitoring their size and zeta-potential over several days. DLS results (Figure 6) show that the α-glucan nanoparticle had a mean diameter of approximately 118 nm after homogenization and increased in size to nearly 122 nm in about 225 days. The error bars indicate little difference in the nanoparticle sizes during this time; however, ANOVA analysis shows that, overall, there was a significant difference (*p* < 0.5), suggesting that the nanoparticles are so stable that their size changed only slightly during the 7 months and that they do not aggregate. It should be noted that the filtration of the α-glucan nanoparticles using a 0.45-micron filter to remove any unusually large nanoparticles did not change the mean diameter (data not shown). Zeta potential measurements show that the nanoparticles had a low zeta potential (~−5 mV) and that it was relatively constant over the same 225 days. Typically, colloidal systems are considered highly stable when they have zeta potential at or above ±30 mV, which is where nanoparticles are less likely to aggregate [18]. The size stability here suggests that despite the low zeta potential, the α-glucan nanoparticles produced by homogenization did not aggregate and were quite stable. This is similar to the findings for similar surfactant-free nanoparticles made from enzymatically synthesized inulin [19].

It is also important to monitor the stability of nanoparticles as a function of pH levels [20,21,22]. Figure 7 shows the size and zeta potential results for nanoparticles created at three different pH levels (pH 5.5, 7.4, and 10). Buffering agents MES, TES, and CHES were used to maintain the respective pH levels of 5.5, 7.4, and 10 of the nanopore water in which the α-glucans were hydrated. α-Glucans homogenized to form nanoparticles at pH 5.5 had a mean hydrodynamic diameter of nearly 120 nm; those nanoparticles that formed at pH 7.4 were slightly larger, approximately 122 nm in diameter. α-Glucan nanoparticles that formed at pH 10, however, had a mean diameter of approximately 115 nm. Statistical analysis using a one-way ANOVA test showed that these sizes were significantly different. The different sizes of α-glucan nanoparticles suggest that nanoparticles created under acidic and neutral conditions are an optimal size, and those made under basic conditions will obtain their smallest size. Analysis of the zeta potential of the α-glucan nanoparticles under different pH conditions exhibited a decreasing trend in the presence of sulfonic acid buffering agents (−0.3 mV under acidic conditions, nearly −0.8 mV at neutral conditions, and approximately −1.3 mV under basic conditions). It should be noted that without a buffering agent present, the α-glucan nanoparticles had a zeta potential in the range of −3 to −8 mV. This suggests that buffering agents of different chemical structures may influence the surface charge of the nanoparticles, possibly by a shielding effect. This is to be further evaluated in later studies.

It is also important to understand the stability of the α-glucan nanoparticles under various temperature conditions. Therefore, accelerated exposure was conducted to determine how varied temperatures affected the nanoparticles. Nanoparticles were exposed to 25 °C (ambient temperature), 37 °C (body temperature), and 60 °C (high temperature) for 5 ½ h intervals, consecutively. Figure 8 shows that exposure to a temperature of 25 °C resulted in the nanoparticle size being relatively unchanged for 8 h; however, the zeta potential linearly decreased to −26 mV, and it is unclear why this was the case. Two hours into the second time interval, where the temperature was 37 °C, the nanoparticles exhibited an exponential increase in average diameter, reaching approximately 119 nm. The nanoparticles exhibited their maximum negative zeta potential of approximately −55 mV, which increased linearly to about −47 mV. Increasing the temperature to 60 °C resulted in the nanoparticles rapidly returning to 108 nm diameter and increasing to approximately 119 nm in diameter over the first half an hour at 60 °C. The nanoparticles then returned to 108 nm in size over the second 30 min period at 60 °C, only to increase in size to 119 nm over the final hour at 60 °C. The zeta potential, on the other hand, appears to stabilize at approximately −39 mV during the entire duration of the nanoparticles being heated at 60 °C. Finally, returning the temperature to 25 °C resulted in apparently immediate shrinkage of the nanoparticles to approximately 99 nm in size. The nanoparticles returned to 108 nm in size over the 30 min period it took for the temperature to equilibrate to 25 °C. The zeta potential, during the 30 min that the temperature was gradually lowered back to 25 °C, decreased to approximately −65 mV and returned to −39 mV once 25 °C was achieved.

## 3. Materials and Methods

### 3.1. Materials

The high-pressure homogenizer, model C5, was purchased from Avestin, Inc. (Ottawa, ON, Canada). 1,6-Diphenyl-1,3,5-hexatriene (DPH) was purchased from ThermoFisher Scientific (St. Louis, MO, USA). Buffering agents 2-(*N*-morpholino)ethanesulfonic acid (MES), 2-[[1,3-dihydroxy-2-(hydroxymethyl)propan-2-yl]amino]ethanesulfonic acid (TES), and 2-(cyclohexylamino)ethanesulfonic acid (CHES) were purchased from Sigma-Aldrich (St. Louis, MO, USA) at chemical grade. A Barnstead NANOpure Diamond UV ultrapure water purification system with a resistivity of 18.2 M Ω cm was the source of the water used throughout this work.

### 3.2. Methods

#### 3.2.1. α-Glucan Synthesis and Structural Characterization

Synthesis and structural characterization of the α-glucan used in this work was carried out previously by Côté and Skory, 2012 [23]. Briefly, the insoluble α-glucans were synthesized using a glucansucrase cloned from the NRRL B-1118 (ATCC 8293) strain of *Leuconostoc mesenteroides*, a lactic acid bacterium, and structurally analyzed using hydrolysis, gas-liquid chromatography, and methylation. Together, these analyses indicated that the glucans had nearly equal proportions of α-(1→3) and α-(1→6) linkages; branching was indicated to have minimally occurred. The final polysaccharide product was freeze-dried and stored until usage.

#### 3.2.2. Differential Scanning Calorimetry (DSC)

Experiments using DSC were explored to ascertain the thermal properties of the α-glucans prior to their formation as nanoparticles. Experiments were conducted in a TA Instruments Q2000 MDSC (TA Instruments, Inc., New Castle, DE, USA). Samples were weighed (~3 mg) and sealed under nitrogen in a hermetic aluminum DSC pan; an empty pan sealed under nitrogen was used as the reference. Samples were equilibrated to 4 °C and heated to 120 °C at a heating rate of 5 °C/min.

#### 3.2.3. Thermogravimetric Analysis (TGA)

Further thermal analysis was conducted by obtaining TGA thermograms using a Q500 TGA instrument (TA Instruments, New Castle, DE, USA). Samples (~10 mg) were placed onto a tared, open platinum TGA pan and heated from room temperature (20–25 °C) up to 600 °C. This was done under nitrogen and at a rate of 5 °C/min. Additional to measuring weight loss, differential TGA weight loss (DTG, %/°C) was also recorded; measurements were conducted in duplicate.

#### 3.2.4. Nanoparticle Preparation via High-Pressure Homogenization

Dried water-insoluble glucan was added to room-temperature water (40 mL) to give a weight-to-volume (*w*/*v*) solution with up to 5% solid content (maximum material passable through the homogenizer). A concentration of 0.25% *w*/*v* (100 mg) was chosen for these experiments and mixed overnight for approximately 18 h to maximize dispersion and hydration. Using an Avestin Emulsiflex-C5 high-pressure homogenizer, the dispersed water-insoluble glucans were homogenized at approximately 70 MPa and continuously passed through the homogenizer 60 times until a clear solution was achieved (conditions determined from evaluating pressures of 35, 70, and 200 MPa and 10–60 passes; see Figure 3b) [24].

#### 3.2.5. Morphological Analysis

The shape of the nanoparticles was determined via scanning electron microscopy (SEM) utilizing a JEOL JSM-6010A (JEOL USA, Inc., Peabody, MA, USA,). All samples were dried, adhered to aluminum specimen mounts by conductive carbon tape, and then sputter-coated with gold.

#### 3.2.6. Fluorescence Anisotropy Measurements

Hydrophobic properties and possible temperature phase transition points of the α-glucan nanoparticles were explored using the fluorescent properties of 1,6-diphenyl-1,3,5-hexatriene (DPH). DPH is well known for partitioning into the hydrophobic region of phospholipid membranes, where the fluorescence of DPH is unquenched compared to DPH residing within water and is used to monitor temperature-induced phase transitions within the lipid bilayer [15,16]. A similar study was undertaken with formed nanoparticles, first to confirm the existence of an internal hydrophobic region (vital for nanoparticles to remain in aqueous solution without separation) and, second, to explore whether the nanoparticles exhibited any detectable temperature-induced phase transitions. DPH in methanol was added to preformed nanoparticles in water; the DPH final concentration in solution was 1 μM. Nanoparticles with DPH were incubated with periodic shaking for 30 min to maximize DPH partitioning into the hydrophobic region of the nanoparticles. DPH/nanoparticle samples were placed into cuvettes and equilibrated to 4 °C. Excitation and emission wavelengths were set to 350 nm and 428 nm, respectively, on a Horiba Jobin-Yvon Fluorlog 3-21 fluorometer (Piscataway, NJ, USA). Anisotropy measurements were conducted from 4 °C to 80 °C at 4-degree intervals; samples were equilibrated at each temperature for 10 min prior to measurements being made, and the sample chamber was purged with pure nitrogen during the entire experiment to eliminate condensation on the cuvettes. Anisotropy was calculated as reported in Evans and Compton, 2017 [5]; the correction factor G was determined by using nanoparticles without DPH but incubated 30 min with the appropriate amount of methanol. Experiments were conducted in triplicate.

#### 3.2.7. Dynamic Light Scattering (DLS) and Zeta Potential Measurements

The size and zeta potential of the nanoparticles were determined using DLS and electrophoresis, respectively, from a Zetasizer Nano-ZS (Malvern, UK) that used 633 nm, 4 mW He-Ne red laser. Measurements were conducted at 25 °C using a 173° detection angle. One measurement entailed up to 13 runs, each lasting up to 20 s per run; 3 runs were conducted per sample. Data were averaged to obtain average particle size and zeta potential. Results are reported as the average of at least triplicate sample measurements. Nanoparticles were also made at three different pH levels to explore how varying pH may affect particle size and zeta potential. Accelerated exposure studies were conducted using a Wyatt Technology Mobius DLS (Wyatt Technology Co., Goleta, CA, USA) and zeta potential system; measurements were conducted at 25 °C, 37 °C, and 60 °C, respectively, for 6 h per temperature.

## 4. Conclusions

This work describes the thermal properties of water-insoluble α-glucans synthesized via glucansucrases and the thermal stability of nanoparticles formed from them. The α-glucan nanoparticles have been shown to remain stable over a 7-month period and up to 86 °C. These properties make the α-glucan nanoparticles potentially highly stable, long-term controlled-release carriers of hydrophobic bioactives and sensors for analytes like mycotoxins that exhibit environment-dependent modulation of fluorescence.

## Figures and Tables

**Figure 1 molecules-25-03807-f001:**
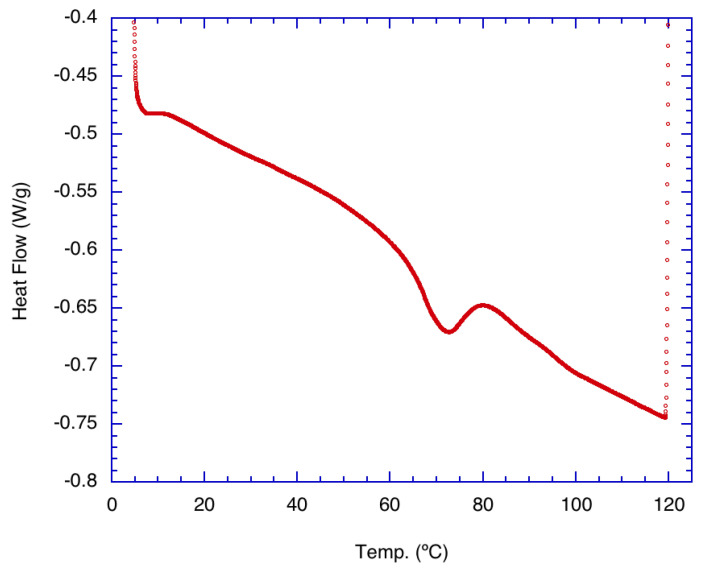
Differential scanning calorimetry (DSC) thermal analysis of purified, freeze-dried alpha glucan synthesized from *Leuconostoc mesenteroides* NRRL B-1118 glucansucrase.

**Figure 2 molecules-25-03807-f002:**
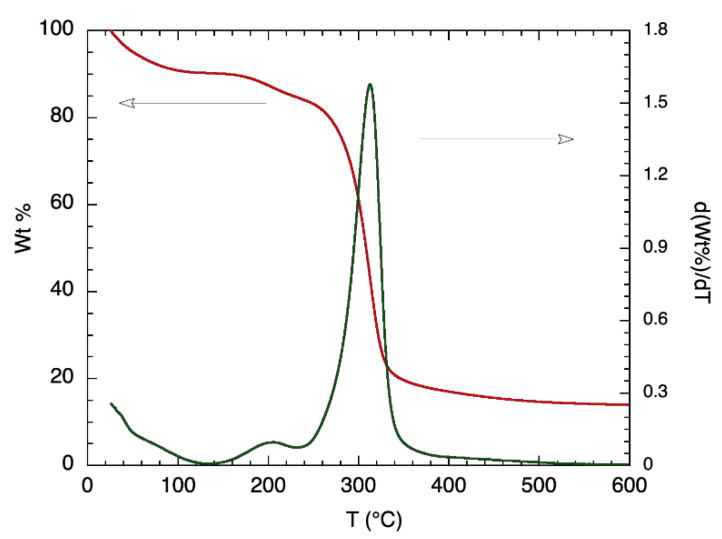
TGA (red line; left y-axis) and differential TGA (green line; right y-axis) thermogram of purified alpha-glucan polysaccharide glucansucrase.

**Figure 3 molecules-25-03807-f003:**
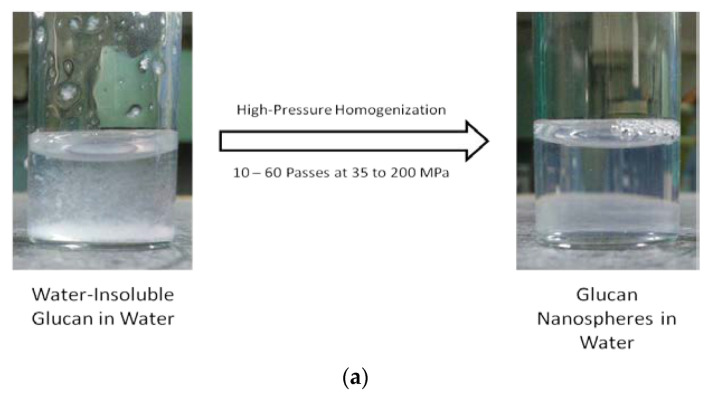
α-Glucans were homogenized in water to clarity to form nanoparticles (**a**). Nanoparticle size as a function of pressure and passes through homogenization (**b**).

**Figure 4 molecules-25-03807-f004:**
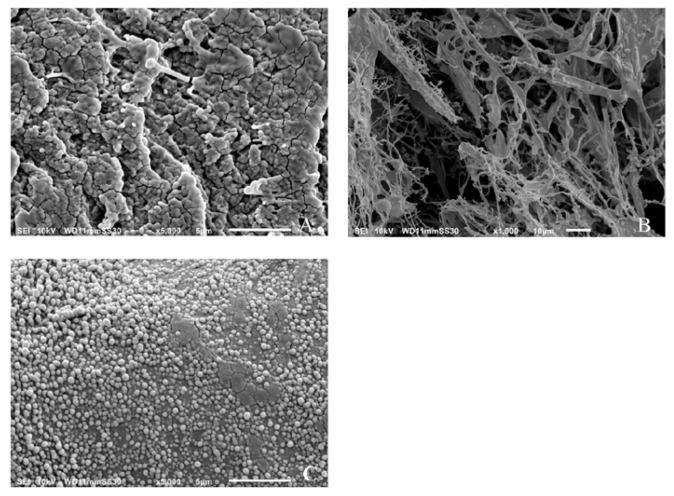
Scanning electron micrograph images of air-dried (**A**), freeze-dried (**B**), and freeze-dried/homogenized (**C**) α-glucans.

**Figure 5 molecules-25-03807-f005:**
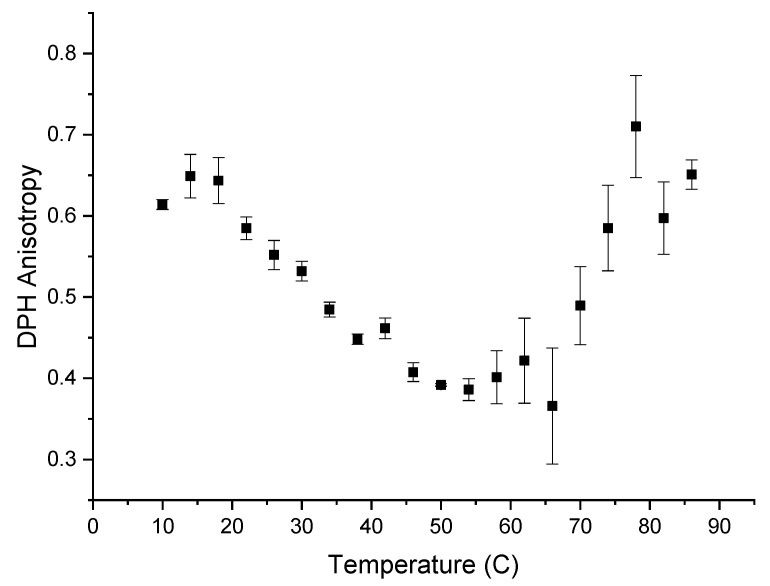
Anisotropy measurement of 1-6-diphenyl-1,3,5-hextriene (DPH) that was intercalated into α-glucan nanoparticles.

**Figure 6 molecules-25-03807-f006:**
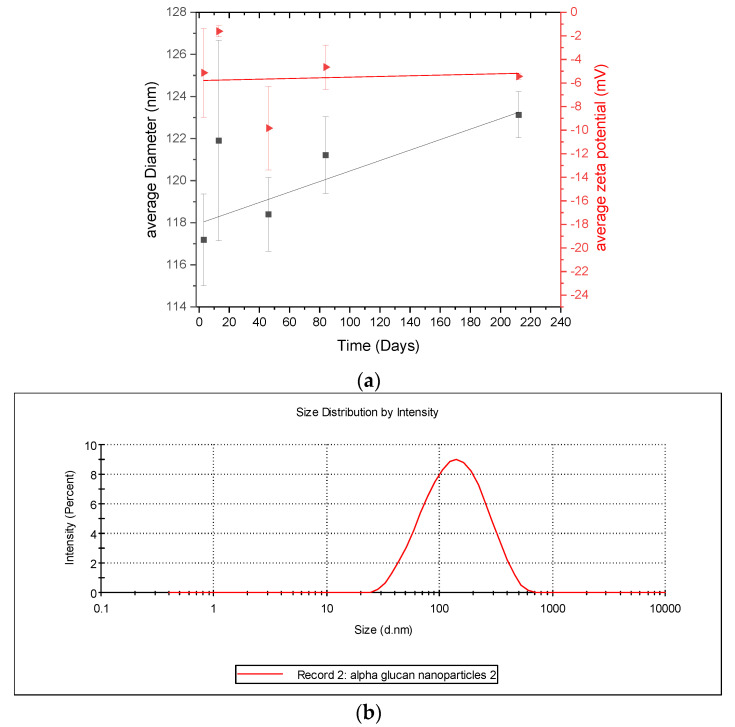
(**a**) Diameter and zeta potential of α-glucan nanoparticles monitored over time; (**b**) representative graph of size distribution of α-glucan nanoparticles during size measurements.

**Figure 7 molecules-25-03807-f007:**
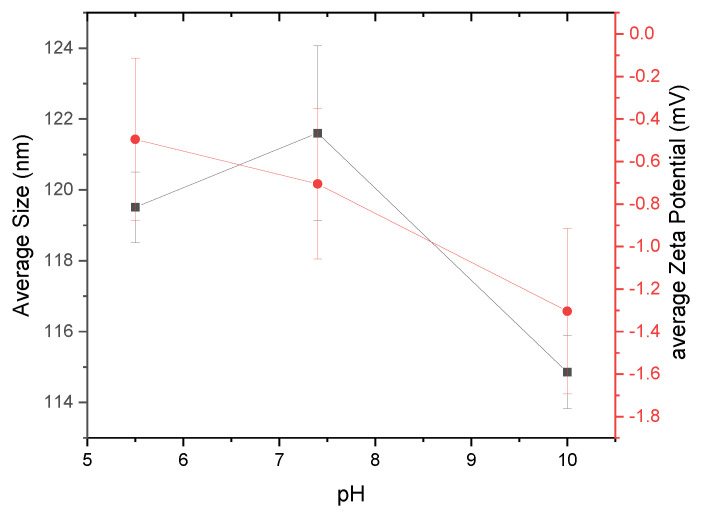
α-Glucan nanoparticles size (black square) and zeta potential (red circle) as a function of pH.

**Figure 8 molecules-25-03807-f008:**
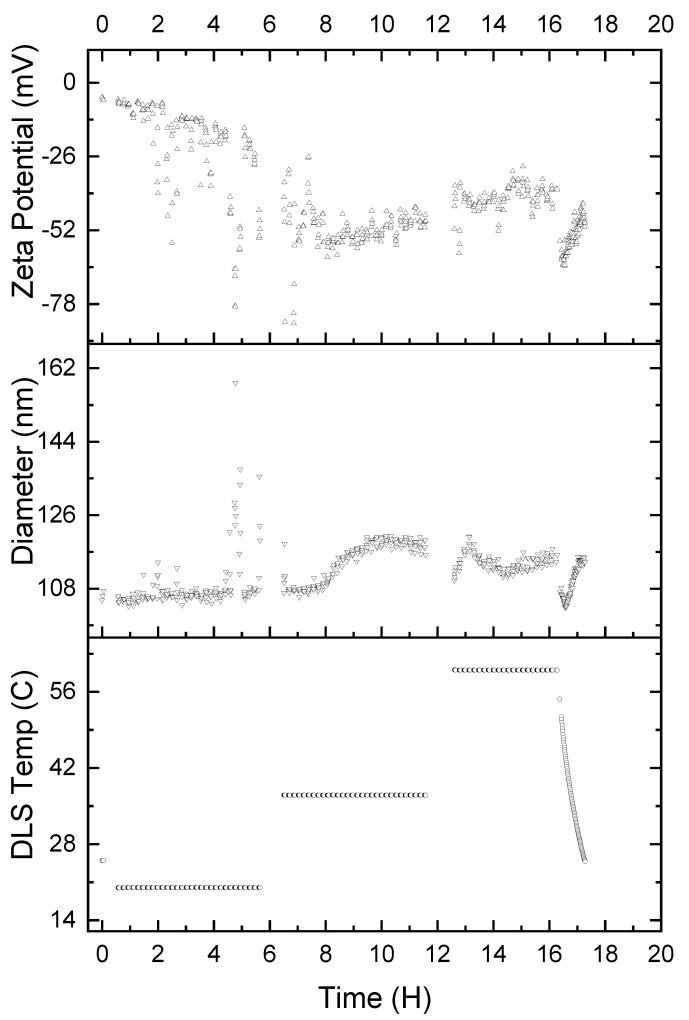
Accelerated exposure of α-glucan nanoparticles for 5 ½ h intervals at 25, 37, and 60 °C, respectively; temperature conditions were gradually returned to 25 °C after 16 h.

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
