# Peer review of "Development and Physical Characterization of α-Glucan Nanoparticles"

_molecules, 2020, doi:10.3390/molecules25173807_

Round 1
Reviewer 1 Report
Reviewer #1: molecules-882407 „Development and Physical Characterization of -Glucan Nanoparticles” Kervin Evans et al.
This manuscript describes the conversion of -glucans into nanoparticles (NPs) of ~ 120 nm in diameter with a surface potential of ~ -3 mV. Using high-pressure homogenization the -glucans were converted into NPs. Glucans were enzymatically synthesized from sucrose using glucan sucrase cloned from Leuconostoc mesenteroides (NRRL B-1118). The authors utilize Fluorescence measurements using 1,6-diphenyl-1,3,5-hexatriene to indicate that the -glucan nanoparticles have a hydrophobic core. Furthermore, the -Glucan nanoparticles were found to be stable for over a 7-month period (even up to 86ºC) and able to form at three pH levels (pH 5.5, 7.4 and 10). These properties make the -glucan nanoparticles potentially highly stable. The results are interesting, but the study lacks a thorough analysis of the sample (the -Glucan nanoparticles) results. Are the sample results statistically significant or not? (p-value) How do the authors determine if a difference is statistically significant?
Thus, minor improvement should be made to the manuscript's current state, prior to it being warranted for publication. The choice of the journal is correct and such a work should be published in Molecules.
Minor typos (I note some of them, but authors should check the manuscript for completeness):
(i) page 1, Abstract, lines 25 and 79: ‘… 220 day …’,
(ii) page 2, Introduction, line 52: ‘… Polysacchide…’,
(iii) page 8, Results and Discussion, line 205: ‘It is also important to monitor stability of nanoparticles are a function of pH levels 21-23’.
Reviewer 2 Report
In this work, the authors presented thermal properties and thermal stability of α-glucans and α-glucans nanoparticles, respectively. The following minor suggestions should be addressed before the manuscript can be considered further.
- It is not clean in the manuscript if the result presented in Figure 1 is for the air dried or freeze-dried α-glucans;
- Line 134- "...may be the degradation temperature for nanoparticles..." Did the authors perform any thermogravimetry (TGA) study with the material? Since one of the goals of the work is to study the thermal properties, TGA would contribute to the work;
- Line 151 - "...similar in size to that determined by DLS measurements" How the author calculated the size distribution of the nanoparticles by SEM? What is the value? The methodology and/or software are not described in the experimental or results section;
- Line 180 - I believe the author wanted to say " Figure 1" in the sentence;
- Did the nanoparticles suspension show a bimodal or multi-modal distribution? It would be informative if the authors could show some DLS curves (and plotted as Intensity vs particle distribution).
Reviewer 3 Report
K. O. Evans et al draft requires major revisions. The method used from synthesis should be described with amounts (even if previous literature is used as reference).
The DSC measuremts for natural based products should made with a maximum heating rate of 5ºC/min.
Also, the authors mention biodegradable in the keywords, and there is no mention in the entire document about that important topic. It should be added.
Considering the classification of nanoparticle size, this work does not produce nanoparticles. Therefore, this should be removed from the draft.
Several characterization tests were performed and simply discussed. When reading the conclusions, the authors wrote "These properties make the -glucan nanoparticles potentially highly stable long-term controlled-release carriers of hydrophobic bioactives and sensors for analytes that exhibit environment dependent modulation of fluorescence", but during the discussion, the values obtained are not cleary link to these applications. A revision of the discussion should be made.
Reviewer 4 Report
Evens and co-workers report on the synthesis of alpha-glucans using high-pressure homogenization. They further reveal the hydrophobicity of the particles using Fluorescence measurements and stability of the particles at various time, temperature, and pH. This is an interesting work and can be accepted after addressing minor points.
- The authors state that depending upon the applied high-pressure homogenization, the size of the glucan nanoparticles can be varied (10-60 passes at 35-200 MPa). In this regard, a plot depicting homogenization pressure vs. quantity of the particle is very useful to gather insights on what different particle sizes are expected.
- If possible, the authors should conduct TEM analyses to provide better images so that one could directly correlate the size obtained from DLS values.
Round 2
Reviewer 3 Report
The authors replied and made the necessary corrections. It can be accepted in the present form.